Molecular assessment of Pocillopora verrucosa (Scleractinia; Pocilloporidae) distribution along a depth gradient in Ludao, Taiwan

De Palmas Stéphane 1 2 3
Soto Derek 1 2 3
http://orcid.org/0000-0002-0914-5586 Denis Vianney 4 vianney.denis@gmail.com
Ho Ming-Jay 2 5
Chen Chaolun Allen 2 3 4 acropora.chen@gmail.com
1 Biodiversity Program, Taiwan International Graduate Program, Academia Sinica and National Taiwan Normal University , Taipei , Taiwan
2 Biodiversity Research Center, Academia Sinica , Taipei , Taiwan
3 Department of Life Science, National Taiwan Normal University , Taipei , Taiwan
4 Institute of Oceanography, National Taiwan University , Taipei , Taiwan
5 Green Island Marine Research Station, Academia Sinica , Ludao, Taitung County , Taiwan
Banaszak Anastazia
Electronic publication date: 2018 Oct 25
Publication date: 2018
Volume: 6
Electronic Location ID: e5797
Received 2018 Aug 7; Accepted 2018 Sep 20
Copyright: © 2018 De Palmas et al.
Copyright year: 2018
Copyright holder: De Palmas et al.
License: This is an open access article distributed under the terms of the Creative Commons Attribution License, which permits unrestricted use, distribution, reproduction and adaptation in any medium and for any purpose provided that it is properly attributed. For attribution, the original author(s), title, publication source (PeerJ) and either DOI or URL of the article must be cited.
License URL: https://creativecommons.org/licenses/by/4.0/

Keywords: Molecular ecology, Deep reef refugia hypothesis, Coral taxonomy, Pocilloporids

Funding: Taiwan International Graduate Program Fellowship Taiwan Ministry of Science and Technology MOST 103-2621-B-001-004-MY3 Derek Soto and Stéphane De Palmas are recipients of the Taiwan International Graduate Program Fellowship. This research was funded by a Taiwan Ministry of Science and Technology Grant number MOST 103-2621-B-001-004-MY3 to Chaolun Allen Chen (https://www.most.gov.tw/?l=en). The funders had no role in study design, data collection and analysis, decision to publish, or preparation of the manuscript.

==============================
It can be challenging to identify scleractinian corals from the genus Pocillopora Lamarck 1816 in the field because of their large range of inter- and intra-specific morphological variation that co-occur with changes in the physical environment. This task is made more arduous in the context of a depth gradient, where light and water current could greatly affect the morphology of the corallum. Pocillopora verrucosa (Ellis & Solander 1786) in Taiwan was previously reported exclusively from shallow waters (<10 m in depth), but a recent observation of this species in the mesophotic zone (>40 m in depth) questions this bathymetric distribution. We used the mitochondrial open reading frame and the histone 3 molecular markers to investigate the vertical and horizontal spatial distribution of P. verrucosa around Ludao (Green Island), Taiwan. We genotyped 101 P. verrucosa-like colonies collected from four depth zones, ranging from 7 to 45 m, at three sites around the island. Of the 101 colonies sampled, 85 were genotyped as P. verrucosa, 15 as P. meandrina, and one specimen as an undescribed Pocillopora species. P. verrucosa was found at all depths, while P. meandrina and the undescribed Pocillopora specimen were limited to 15 m depth. P. verrucosa has a large bathymetric distribution around Ludao and could benefit from the refuge that the mesophotic zone offers. This study illustrates the difficulty of identifying Pocillopora corals in the field and emphasizes the relevance of molecular taxonomy as an important and complementary tool to traditional taxonomy for clarifying vertical and horizontal species distribution. Our results also illustrate the need in conservation biology to target species genetic diversity rather than just species diversity.

Introduction

Scleractinian coral species identification is traditionally based on the classification of coral skeletal features, particularly colony macro-morphology and corallite micro-structure (Veron & Pichon, 1976; Wallace, 1999; Budd et al., 2010). However, some scleractinian species can exhibit environmentally correlated variations in morphology, i.e., ecomorphs (Veron & Pichon, 1976), which often makes species identification a challenge (Todd, 2008; Veron, 2013). This problem is exacerbated when attempting to identify coral species directly in the field; it also highlights the need to redefine species boundaries in light of molecular approaches. In this regard, quantitative morphological and molecular analyses have been applied to delineate species within the genera Acropora (Wallace, 1999), Orbicella (Medina, Weil & Szmant, 1999), Montipora (Van Oppen, Koolmees & Veron, 2004), Platygyra (Mangubhai, Souter & Grahn, 2007), Pocillopora (Flot et al., 2008), Seriatopora (Chen et al., 2008), Porites (Forsman et al., 2009), Psammocora (Benzoni et al., 2010) and Stylophora (Keshavmurthy et al., 2013).

Past taxonomic studies have reported up to 35 Pocillopora ecomorphs (see Veron & Pichon, 1976). However, facing a large spectrum of morphological variations at both the intra- and inter-species level, various studies have hypothesized that the actual number of species in this taxon could be overestimated (Veron & Pichon, 1976; Veron, 2013). For example, Pocillopora colonies may display morphology corresponding to other ecomorphs when transplanted into different environmental conditions (Lesser et al., 1994; Hoogenboom, Connolly & Anthony, 2008; Prada, Schizas & Yoshioka, 2008; Todd, 2008; Paz-García et al., 2015) or exhibit morphological plasticity along a depth gradient (Soto et al., 2018). In the last decade, a growing body of literature has focused on resolving the taxonomy of Pocillopora by assessing morphological traits in conjunction with genetic markers (Flot et al., 2008; Pinzón & LaJeunesse, 2011; Pinzón et al., 2013; Schmidt-Roach et al., 2012, 2014; Marti-Puig et al., 2014). These studies have identified the mitochondrial open reading frame (mtORF) as an efficient marker for delineating Pocillopora species. The mtORF marker has been recently used by Johnston et al. (2017) in conjunction with a genus-wide genomic comparison of Pocillopora, which confirmed it as a suitable and fast tool for delineating most Pocillopora species.

The mtORF marker has been used to revisit the taxonomy of Pocillopora, thus addressing species overestimation by consolidating synonymous species. According to Schmidt-Roach et al. (2014) and based on the mtORF marker, the Pocillopora genus is divided into five genetic lineages (or clades), each containing one to three closely related species. Clade 1 is composed of Pocillopora damicornis, P. acuta and P. aliciae. Clade 2 consists of P. verrucosa and the recently described P. bairdi. Clade 3 is represented by P. meandrina and P. eydouxi, which share the same mtORF but could be further distinguished based on the histone 3 (hist 3) region (Johnston, Forsman & Toonen, 2018). P. cf. brevicornis is the sole member of clade 4, while P. ligulata and P. cf. effusus compose clade 5. Finally, an undescribed Pocillopora sp. (type 8) has an unclear position within the Pocillopora phylogeny but is considered as a valid taxon (Flot et al., 2008; Pinzón et al., 2013; Schmidt-Roach et al., 2014). The mtORF and hist 3 regions are the major molecular tools for assessing the Pocillopora distribution in the Indo-Pacific region (Gélin et al., 2017a, 2017b; Poquita-Du et al., 2017; Johnston, Forsman & Toonen, 2018; Torres & Ravago-Gotanco, 2018). However, while most studies have focused on the horizontal distribution of Pocillopora (Gélin et al., 2017a, 2017b; Poquita-Du et al., 2017; Johnston, Forsman & Toonen, 2018), its distribution along a vertical gradient has not been specifically addressed, despite mentions of its presence at various depths (Ziegler et al., 2014; Gorospe & Karl, 2015).

Pocillopora species are found in most reefal and non-reefal coral communities surrounding Taiwan and its offshore islets. P. verrucosa, P. meandrina, P. damicornis and P. eydouxi have been described as exclusively shallow water species, their distribution ranging from 0 to 15 m depths (Dai & Horng, 2009a). However, Denis et al. (in press) reported P. verrucosa in Ludao (Green Island) at depths of up to 55 m, where it is one of the dominant scleractinian coral. Locally, P. meandrina and P. verrucosa share close morphologies: P. meandrina is described as similar to P. verrucosa but with shorter, flattened branches and smaller verrucae (Dai & Horng, 2009b). Both species are sympatric in Taiwanese waters, with colonies of both species sometimes found next to each other. Due to the effects of environmental plasticity, the corallum macromorphology is not considered as a diagnostic character in the Pocillopora genus (Paz-García et al., 2015; Gélin et al., 2017b) and species identification in the field could easily be confused. Therefore, we proposed to re-investigate horizontal and vertical distribution of P. verrucosa around Ludao using a molecular approach. This molecular assessment is essential to estimating the overall biodiversity in the mesophotic zone as well as for estimating the degree of overlap between shallow and mesophotic communities. The latter is of critical importance to decision making for conserving targeted species.

Material and Methods

Selected sites and sampling

Three sites around Ludao, off the southeastern coast of Taiwan, were selected for this study: Guiwan, Dabaisha and Gongguan (Fig. 1). Guiwan was surveyed in 2016 and Dabaisha and Gongguan were surveyed in 2017. At each site, large fragments of at least five colonies tentatively identified as P. verrucosa were collected from 7, 15, 23–30 to 38–45 m in depth (tidal amplitude ± 1.5 m). Subsamples were collected from each fragment and preserved in 90% ethanol for molecular analysis, and the remaining skeletons were bleached and dried for morphological observation. Coral tissue samples were collected under Taitung County Government permit number 1040000285.

Figure 1 Map of Taiwan showing location of Ludao and sampling locations.

(A) Taiwan settings and the position of Ludao; (B) details of Ludao and position of the three sites selected in this study.

DNA extraction

Small subsamples were ground and homogenized in 250 μL of SDS lysis buffer (1M Tris–HCl, 5M EDTA, 20% SDS, 5M NaCl, pH 8) and incubated at 57 °C for 12 h with Proteinase K (Sigma-Aldrich, Saint-Louis, MO, USA) at a final concentration of 10 μg mL−1. DNA was extracted using Phenol:Chloroform:Isoamyl alcohol (25:24:1, Sigma-Aldrich, Saint-Louis, MO, USA) and precipitated using ethanol (−20 °C). The precipitates were washed in 70% ethanol (−20 °C) and DNA pellets were dried at room temperature before being re-suspended in 100 μL of sterile TE buffer 1× (USB Corporation, Cleveland, OH, USA).

Molecular analysis

The mtORF region was amplified using “FATP6.1” (5′-TTTGGGSATTCGTTTAGCAG-3′) and “RORF” (5′-SCCAATATGTTAAACASCATGTCA-3′) primers following the protocol described in Flot et al. (2008). Polymerase Chain Reaction (PCR) mixes contained 20 μL of Master Mix RED (Ampliqon, Odense M, Denmark) mixed with 15 μL of ddH2O, two μL of each forward and reverse primer (2.5 μM) and two μL of template DNA (5–50 ng). The PCR consisted of a 60 s denaturation step at 94 °C, followed by 40 cycles of 30 s at 94 °C, 30 s at 53 °C and 75 s at 72 °C, ending in an extension step at 72 °C for 5 min. PCR products were sequenced in both directions using an ABI 3730XL system (Thermo Fisher Scientific Inc., Waltham, MA, USA). Sequences were manually edited using SeqMan (Lasergene Sequence Analysis Software, Madison, WI, USA) and aligned using the ClustalW algorithm implemented in MEGA 7 (Kumar, Stecher & Tamura, 2016). Sequence files were converted to the Phylip format for analysis in PopArt (Leigh & Bryant, 2015). The Median-Joining Network method was used to illustrate the relationship among recovered sequences (Bandelt, Forster & Röhl, 1999).

All samples that cluster with clade 3 may belong either to P. eydouxi or P. meandrina. To further differentiate both species, the hist 3 region was amplified using PocHistoneF: 5′-ATTCAGTCTCACTCACTCACTCAC-3′ and PocHistoneR: 5′-TATCTTCGAACAGACCCACCAAAT-3′ primers following the protocol described in Johnston, Forsman & Toonen (2018). The same PCR mix and amplification program described above for mtORF was used for hist 3. PCR products were used in restriction fragment length polymorphism (RFLP) analysis and digested using Xho I restriction enzyme (Thermo Fisher Scientific Inc., Waltham, MA, USA) following the manufacturer’s recommendations. A total of 10 μL of digested products were electrophoresed on 2% agarose gel at 100 V for 25 min. The gel was photographed using a Molecular Imager XR+ (Biorad, Hercules, CA, USA). Digestion of the hist 3 region by Xho I distinguishes P. eydouxi (two digestion products at ∼287 and ∼382 bp) and P. meandrina (one single digestion product at ∼669 bp), as the restriction site is absent in P. meandrina (Johnston, Forsman & Toonen, 2018).

Results

From haplotype to species diversity

A total of 101 mtORF sequences were analyzed in this study, representing a total of nine haplotypes (H1–H9). Haplotypes H1 (n = 1) and H2 (n = 14) clustered with clade 3 (Fig. 2), which comprises P. eydouxi and P. meandrina. The hist 3 PCR–RFLP revealed that all 15 samples (H1 and H2) belong to P. meandrina (Fig. S1). Haplotype H3 (n = 1) matched Pocillopora Type 8a, an undescribed Pocillopora species (Fig. 2). Haplotypes H4 (n = 1), H5 (n = 2), H6 (n = 3), H7 (n = 9), H8 (n = 10) and H9 (n = 60) all matched published references of P. verrucosa from clade 1 (Fig. 2). These haplotypes (H4–H9) have up to six base pairs differences with each other (Fig. 2), and represent 84% of the total genotyped colonies. The genetic relationship between all haplotypes is illustrated in the haplotype network (Fig. 2).

Figure 2 Haplotype network based on the mtORF sequence data recovered in this study (total alignment length 656 bp).

Vertical bars represent the number of base pair differences between haplotype. Pocillopora clade 2 is blue, Pocillopora type 8 is brown, Pocillopora clade 3 is red.

Haplotype bathymetric distribution

Haplotypes H1 and H2 originated from 7 and 15 m. Haplotype H3 was found at seven meters. These three haplotypes belong to P. meandrina and an undescribed Pocillopora species, neither of which were targeted in this study. Haplotypes H4–H9 were genotyped as P. verrucosa and were found distributed as follows: Haplotypes H4 was found at 23–30 m in depth and H5 was found at both 23–30 and 38–45 m. Haplotype H6 was found at intermediate depth ranges, 15 and 23–30 m, and haplotypes H7 and H9 were found at all depths while H8 was found at all depths except 15 m (Fig. 2).

Discussion

This study found nine haplotypes that correspond to three Pocillopora species while targeting “typical” P. verrucosa morphologies (Dai & Horng, 2009b). Of the 101 colonies sampled, 85 (84%) clustered with P. verrucosa (clade 2, Fig. 2), and match published mtORF references of this species (Schmidt-Roach et al., 2012; Pinzón et al., 2013; Hsu et al., 2014; Marti-Puig et al., 2014; Gélin et al., 2017b). A total of 15 (15%) clustered with the complex of species P. meandrina and P. eydouxi (clade 3, Fig. 2) and were further identified as P. meandrina after hist 3 PCR–RFLP analysis (Fig. S1). Finally, one specimen (1%) matched Pocillopora sp. (type 8a, Pinzón et al., 2013), an undescribed Pocillopora species (Fig. 2). Our results show that genotypes belonging to different species may be confounding because they have the same apparent morphology (i.e., “ecotype,” see Fig. S2). Recently, Gélin et al. (2017b) reached a similar conclusion when studying a large Pocillopora collection spanning from the western Indian Ocean to the central Pacific Ocean. They found either a single haplotype displaying morphological characteristics of several morpho-species, or a single morpho-species harboring different haplotypes. Johnston, Forsman & Toonen (2018) showed comparable results in Hawaii: out of 691 coral fragments displaying a P. meandrina-like morphology, 222 were P. ligulata. In the same study, 24 out of 25 samples presenting a P. damicornis-like morphology were genotyped as P. acuta in Kaneohe Bay, Hawaii. Interestingly Pocillopora from the same location have been previously referred to as P. damicornis (Mayfield et al., 2010; Gorospe & Karl, 2013; Putnam & Gates, 2015). In Singapore, P. damicornis-like specimens were genotyped as P. acuta by Poquita-Du et al. (2017). As highlighted in previous studies, our results demonstrate the limitations of using morphology alone to identify Pocillopora in the field (Flot et al., 2008; Pinzón et al., 2013; Schmidt-Roach et al., 2012, 2014; Johnston, Forsman & Toonen, 2018) and emphasize the relevance of molecular taxonomy in supporting studies on the biology and ecology of Pocillopora species.

We found all six P. verrucosa haplotypes (H4–H9) referenced in past studies (Table 1). Interestingly, haplotypes H4, H5 and H6 were initially reported exclusively in the Red Sea and the Arabian Gulf and were previously believed to be regionally endemic (Pinzón et al., 2013). However, they have been identified at other locations such as Reunion Island in the Indian Ocean and New Caledonia in the Pacific (Gélin et al., 2017b). Their presence in Taiwan constitutes a considerable extension of their biogeographic range to northern latitudes, and suggests that they may be much more cosmopolite than previously thought. Yet, they were only found at a low frequency around Ludao since H4, H5 and H6 represent 1%, 2% and 4% of the genotyped P. verrucosa colonies, respectively. Haplotype H4 was found at 23–30 m in depth and H5 was found at 23–30 and 38–45 m, haplotype H6 was found at intermediate depth ranges (15 and 23–30 m). These haplotypes (H4–H6) seem to be present from intermediate to deep habitats, but this is speculative given their low frequency in our results. Moreover, these haplotypes could harbor morphology that significantly differs from typical P. verrucosa morphology in shallow waters. We therefore recommend extending the sampling efforts to other Pocillopora ecotypes in future assessments of the Pocillopora diversity around Ludao. Haplotypes H7, H8 and H9 have a large distribution throughout the Indo-Pacific region (Table 1). They represent the dominant P. verrucosa haplotypes found in this study as they count for 93% of the genotyped P. verrucosa colonies. They were also previously found in coral recruits in the shallow waters of Kenting, southern Taiwan (Hsu et al., 2014). In our study haplotype H8 was not found at 15 m. However, we suspect that this haplotype could be present at this depth given that it was collected at all other depths. H7 and H9 were found at all depths, suggesting that they have a large biogeographic and bathymetric distribution around Ludao. If differences in physiological performance between haplotypes exist, then H7 and H9 could represent generalist lineages able to survive contrasting environmental settings. Aside from examining the genetic diversity of P. verrucosa, this study is the first, to our knowledge, to consider the depth distribution of its haplotypes. There is no evidence that different haplotypes could confer any physiological advantage under contrasting environmental conditions. Therefore, further research is needed into whether the distribution of haplotypes echoes any environmental patterns.

Table 1 Summary of haplotype diversity per site, corresponding to literature references, and their geographic locations previously collected.

Haplotype numbera	Corresponding speciesb	Corresponding haplotype names	Documented location	Sites	
Guiwan	Dabaisha	Gongguan	
H1	Pocillopora meandrina (clade 3)	–	Taiwan	1	–	–	
H2	Pocillopora meandrina (clade 3)	e/m (Schmidt-Roach et al., 2012), Type 1a (Pinzón et al., 2013), clade IIb (Marti-Puig et al., 2014), NA (Hsu et al., 2014), ORF 27 (Gélin et al., 2017b)	Andaman Sea, Clipperton Atoll, Cook Isl. Eastern Australia, Europa Isl, Galapagos, Glorioso Isl., Hawaii, Howland Isl., Johnston Atoll, Juan de Nova Isl., Lizard Isl., Madagascar, New Caledonia, Niihau, Palau, Panama, Phoenix Isl., Reunion Isl., Rodrigues Isl., Taiwan, Tanzania, Tromelin Isl., Zanzibar	5	4	5	
H3	Pocillopora sp. (type 8)	Type 8a (Pinzón et al., 2013), ORF 23 (Gélin et al., 2017b)	Chesterfield Isl., Cook Isl., New Caledonia, Taiwan	–	–	1	
H4	Pocillopora verrucosa (clade 2)	Type 3g (Pinzón et al., 2013), ORF 43 (Gélin et al., 2017b)	Arabian Gulf, New Caledonia, Red Sea, Reunion Isl.	–	–	1	
H5	Pocillopora verrucosa (clade 2)	Type 3h (Pinzón et al., 2013), ORF 35 (Gélin et al., 2017b)	New Caledonia, Red Sea	–	2	–	
H6	Pocillopora verrucosa (clade 2)	Gamma (Schmidt-Roach et al., 2012), Type 3a (Pinzón et al., 2013), ORF 44/ORF 45 (Gélin et al., 2017b)	New Caledonia, Red Sea	2	1	–	
H7	Pocillopora verrucosa (clade 2)	Type 3b (Pinzón et al., 2013), NA (Hsu et al., 2014), Clade IIa (Marti-Puig et al., 2014), ORF 47 (Gélin et al., 2017b)	Chesterfield Isl., Galapagos, Lizard Isl., New Caledonia, Palau, Taiwan, Tonga, Zanzibar, Western Australia	6	–	3	
H8	Pocillopora verrucosa (clade 2)	Gamma (Schmidt-Roach et al., 2012), Type 3f (Pinzón et al., 2013), NA (Hsu et al., 2014), ORF 54 (Gélin et al., 2017b)	Andaman Sea, Eastern Australia, Lizard Isl., New Caledonia, Palau, Taiwan	4	3	3	
H9	Pocillopora verrucosa (clade 2)	Type 3f (Pinzón et al., 2013), NA (Hsu et al., 2014), ORF 53 (Gélin et al., 2017b)	Chesterfield Isl., Lizard Isl., New Caledonia, Taiwan, Tonga	19	25	16	
Notes:

a Haplotypes nomenclature used in this study.

b Corresponding nomenclature following Schmidt-Roach et al. (2014).

This study’s findings broaden our knowledge of the P. verrucosa distribution around Ludao. This species was previously known from shallow waters from 0 to 10 m deep (Dai & Horng, 2009a). Our data corroborate the presence of P. verrucosa in the mesophotic zone of Ludao (Denis et al., in press). This finding extends the known bathymetric distribution of this species in Ludao, with the help of molecular taxonomy. P. verrucosa is one of the dominant scleractinian corals at the maximum depth surveyed by Denis et al. (in press; 55 m in depth) and in our survey as well (45 m in depth). The relatively important density of P. verrucosa at these depths suggests that this species could occur at greater depths than the ones surveyed. In the literature, P. verrucosa is usually considered a very common reef builder in shallow waters but rare below 30 m in depth (Veron & Pichon, 1976). However, several records of this species in the mesophotic zone have been reported (Kühlmann, 1983; Bouchon, 1981), with the deepest record at 54 m (Reyes-Bonilla et al., 2005). Interestingly, Titlyanov & Latypov (1991) found P. verrucosa in habitats where surface irradiance was reduced by more than 95% at 20 m in depth, suggesting that this species can actually be found close to the lower limit of the mesophotic coral ecosystem (MCE) zonation (i.e., where 1% of the surface photosynthetic active radiation remains). With the knowledge accumulating on MCEs, several species previously considered as present only in shallow waters were found in the mesophotic zone. Recently, deep community composition has been shown to overlap the shallow one by 26–97% (57% for Scleractinia), depending on location (Laverick et al., 2018). This information is crucial to understand whether deep water coral assemblages are continuations of the shallow ones or independent. P. verrucosa in Ludao can be considered as a species that contributes to this community overlap, and the next rational step is to understand if the deep populations contribute to the dynamics of the shallow populations.

By specifically targeting typical P. verrucosa morphology, this study cannot be conclusive about the distribution of “bycatch” Pocillopora species. However, the presence of two additional Pocillopora species in our sampling can be informative in regards to the diversity and distribution of those species around Ludao. P. meandrina is represented in our dataset by two haplotypes (H1 and H2). While haplotype H2 is widespread throughout the Indo-Pacific region (Table 1), haplotype H1 is new and distinguished from H2 by one bp. The genotyping of this species exclusively in the shallow waters could indicate that P. verrucosa and P. meandrina are, at least, difficult to differentiate in the shallow waters of Ludao. Our deepest record of P. meandrina genotype was limited to 15 m and this species was not found in the diversity survey done in Denis et al. (in press). Moreover, the undescribed Pocillopora species genotyped in our survey (H3), recovered from the shallow water (seven meters) confirms the presence of this rare Pocillopora species from Taiwan (Pinzón et al., 2013; Schmidt-Roach et al., 2014). Recently this species has also been found inhabiting the shallow waters of Cook Island and New Caledonia (including Chesterfield Island, Gélin et al., 2017b). We propose that both species (P. meandrina and the undescribed Pocillopora species) should receive more attention in future diversity assessments in order to clarify their biogeographic and bathymetric distributions. We emphasize that the diversity of Pocillopora should be revisited in light of recent advances in molecular taxonomy.

The deep reef refugia hypothesis (DRRH, Glynn, 1996; Bongaerts et al., 2010) stipulates that mesophotic habitats (>30 m) could be sheltered from perturbations occurring in shallow waters. Mesophotic coral populations could therefore contribute to the recruitment of shallow water populations, supporting their recovery. The degree of overlap between shallow and deep communities, the fecundity of deeper organisms, and the ability of an offspring from deeper habitats to survive in the shallows, are premises to this hypothesis (Holstein, Smith & Paris, 2016; Laverick et al., 2016; Loya et al., 2016). In this regard, more investigation is needed to clarify the distribution of P. meandrina and the unidentified Pocillopora species with depth. If their presence is confirmed to be restricted in the shallow waters of Ludao, they might not benefit from the deep reef refugia scenario. In contrast, the distribution of P. verrucosa over shallow and mesophotic depth zones demonstrates that this species fulfills at least one criterion of the deep reef refugia recovery scenario. The recovery of this species could potentially then rely on the recruitment of coral larvae from deeper populations as well as from surrounding shallow water populations. If the DRRH usually applies at the species level, it should be expanded to include MCEs as refuges for genetic diversity. Scleractinian coral populations may suffer from genetic loss after perturbations such as bleaching, storms, pollution or diseases. In turn, reduced genetic diversity can result in higher vulnerability of coral populations to these perturbations (Baums, Miller & Hellberg, 2006). The finding of six P. verrucosa haplotypes, with at least three of them which were present along the depth gradient (H7, H8 and H9), could reflect a high genetic diversity of this species around Ludao. Maintaining this genetic diversity along the depth gradient is crucial to ensuring the survival of P. verrucosa when facing the adverse effects of environmental fluctuations and anthropogenic activities. Consequently, species overlap between shallow and deep habitats should not be the only focus of DRRH testing (Rocha et al., 2018). Our results highlight that species distribution and haplotype diversity should be considered in DRRH testing and in conservation decisions. Future investigations on the vertical and horizontal genetic connectivity, fecundity of deeper populations and survival of recruits in shallow water for P. verrucosa around Ludao should be considered in order to further address other DRRH premises. Overall, incorporating a molecular approach, alongside a traditional coral taxonomy one, reduces the risk of misidentification prior to any ecological, physiologic or genetic investigations.

Conclusion

It is particularly difficult in the field to identify species of scleractinian corals that manifest morphological plasticity associated with environmental changes. In this case, a molecular approach is required to correctly and quickly delineate coral species and provide a better understanding of coral species biology and ecology. Here, we show that P. verrucosa can have a wider bathymetric distribution than previously thought in Ludao, Taiwan. Moreover, we found several haplotypes of this species living in sympatry from shallow to deep water. The presence of this species along the depth gradient fulfills the first premise of the DRRH and makes this species of particular interest to evaluate the contribution of shallow and deep populations to recruitment and population maintenance. While molecular approaches have been used to revisit the diversity of major scleractinian taxa, their use has been mostly restricted to horizontal distribution. This study paves the way to investigate vertical distribution by implementing the molecular method.

Supplemental Information

Supplemental Information 1 Histone 3 FASTA file.

Click here for additional data file.

Supplemental Information 2 mtORF FASTA file.

Click here for additional data file.

Supplemental Information 3 Fig. S1. Photograph of histone 3 PCR-RFLP gel electrophoresis.

Lanes 1 and 17 are used for the ladder (bp). Samples are in the following order: a-GI3036, b-GI3037, c-GI3038, d-GI3039, e-GI3040, f-GI3044, g-GI6003, h-GI6004, i-GI6010, j-GI6025, k-GI6056, l-GI6058, m-GI6071, n-GI6075, o-GI6077.

Click here for additional data file.

Supplemental Information 4 Fig. S2. Representative coral fragments of each identified haplotypes.

A–H1 (GI3040), B-H2 (GI6058), C- H3 (GI6057), D- H4 (GI6092), E- H5 (GI3014), F- H6 (GI3034), G- H7 (GI6099), H- H8 (GI3018), I- H9 (GI3003). Numbers in brackets are the sample numbers. Scale bar is 6cm long. Photo credit: Stéphane De Palmas.

Click here for additional data file.

The authors wish to acknowledge the Green Island Marine Research Station and Academia Sinica for logistical assistance. We also want to thank Yeng Su for his amazing assistance with technical diving and sample collection.

Additional Information and Declarations

Competing Interests

Author Contributions

Field Study Permissions

Data Availability

The authors declare that they have no competing interests.

Stéphane De Palmas conceived and designed the experiments, performed the experiments, analyzed the data, contributed reagents/materials/analysis tools, prepared figures and/or tables, authored or reviewed drafts of the paper, approved the final draft, original draft.

Derek Soto performed the experiments, authored or reviewed drafts of the paper, approved the final draft, samples collection.

Vianney Denis conceived and designed the experiments, authored or reviewed drafts of the paper, approved the final draft.

Ming-Jay Ho approved the final draft, samples collection.

Chaolun Allen Chen conceived and designed the experiments, contributed reagents/materials/analysis tools, authored or reviewed drafts of the paper, approved the final draft.

The following information was supplied relating to field study approvals (i.e., approving body and any reference numbers):

Coral tissue samples were collected under Taitung County Government permit number 1040000285.

The following information was supplied regarding the deposition of DNA sequences:

The ORF and the Histone 3 are available as Supplemental Files and all the sequences are available at Dryad: doi:10.5061/dryad.5h01m0c.

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
