# Peer review of "Molecular assessment of Pocillopora verrucosa (Scleractinia; Pocilloporidae) distribution along a depth gradient in Ludao, Taiwan"

_PeerJ, doi:10.7717/peerj.5797_

## Round 0.1 · original submission · Minor Revisions

Two expert reviewers have evaluated your manuscript and their comments can be seen below. As you can see, both reviews are favourable with only minor corrections that need to be made to the manuscript.

Reviewer 1 ·

Basic reporting

The manuscript Molecular assessment of Pocillopora verrucosa (Scleractinia; Pocilloporidae) distribution along a depth gradient in Ludao, Taiwan by De Palmas et al. present basic and important evidence of vertical distribution of different haplotypes belonged to P. verrucosa and other two species with similar morphology. Additionally, this study documents the frequency of misidentification of Pocillopora species according to morphology. Although the manuscript is well organized in sections, also had some issues that should be addressed before publication (see below).

Experimental design

Authors identified Pocillopora species identity using mtORF and RFLP methods of Histone region. Overall the methods and techniques used are suitable for addressing the question they are asking. Although the amplification of ORF is well validated for identification of Pocillopora species through the Indo-Pacific, Histone region has been only tested and validated in Hawaii. I recommend at least have one or two histone sequences from each sample identified by using ORF marker. This will help in future studies comparing Pocillopora species in several regions.

Validity of the findings

This study is particularly important to untangle the ecological differences of Pocillopora species and the method that is described in the manuscript might be replicable in other regions.

Additional comments

Minor comments
Line 71. Most of the recent genetic studied divide the genus Pocillopora in similar genetic lineages and species. However, the denomination of each group change depending of the author. Please specific the publication that you are citing. For example: According to Schmidt-Roach et al. (2014), Clade 1 is composed of . . .
Although the authors indicate the identity of the clades with the haplotypes of ORF in the manuscript. I suggest use the current valid name of the species identified by molecular methods along the manuscript, figures, supplementary material, etc. This will help to avoid confusion in the scientific literature. For example P. verrucosa (Clade 2), complex of species P. meandrina and P. eydouxi (clade 3) and Pocillopora sp. (Type 8).

Line 90-92 Some changes in the colonial and branch morphology are associated to light and high or low current conditions. Please review the following papers: Kaniewska et al. 2008 (Mar Biol 155:649–660) and Paz-García et al. 2015 (Oecologia 178: 207–218).

Line 129 Change incubation by extension.

Line 275-276 Please indicate the meaning of DRRH in the first mention in the text.
Line 289 Please indicate the meaning of MCEs
In figure 2 use name of the Pocillopora species and in parenthesis the number of the clade

·

Basic reporting

no comment (see general comments)

Experimental design

no comment (see general comments)

Validity of the findings

no comment (see general comments)

Additional comments

Review - Molecular assessment of Pocillopora verrucosa (Scleractinia; Pocilloporidae) distribution along a depth gradient in Ludao, Taiwan




General comments: This study examined the depth distribution of Pocillopora in Taiwan, focusing on P. verrucosa- like morphology. The genus is notorious for morphological plasticity, therefore molecular markers (specifically a mitochondrial marker and nuclear marker RFLP assay) were used to confirm species identity and examine haplotype differences across the sampled range. The findings indicated that P.verrucosa is distributed across a broad depth gradient, with no clear genetic differentiation across depths, implying that for this species deeper waters may serve as a refugia. P.meandrina was also rarely misidentified in ~10% of the samples and appear to be more restricted to shallow waters, although additional studies are needed to confirm this. Overall the manuscript is extremely well written, very concise and very well done. It is rare for me to not be able to make suggestions to improve the manuscript, but I think this paper is suitable for publication after some minor suggestions.



Specific comments:


I don’t think the subheadings in [] are necessary. I would leave them out.



Line 76: P. cf. effusus not effuses. Autocorrect always changes it to effuses so watch out.

Line 77: an unclear position into the Pocillopora phylogeny

Line 98: the degree of overlap

Line 108: Subsamples were collected from each fragments

Line 168: which were targeted in this study.. Delete extra period.

Line 169: Haplotypes H4 was found at 23-30m in depth, and H5 was found at 23-30m

Line 204: How about “However, they have since been identified at other locations such as Reunion Island…”

Line 207: more cosmopolite than previously though. Use thought, not though.

Line 219-220: Haplotype H8 was not found at 15m, but we suspect that it could be present at all depth since we found it at all other depths.

This is a little confusing. How about “In our study haplotype 8 was not found at 15m. However, we suspect that this haplotype could be present at this depth given that it was collected at all other depths.”

Line 233: Our data corroborate the presence of P. verrucosa in the mesophotic zone of Ludao (Denis et al. In press).

Line 248: deep community composition has been shown to

Line 251: species that contributes to this community overlap

Line 280: the ability of an offspring from deeper habitats to survive in the shallows, and the fecundity of deeper organisms are premises to this hypothesis.

What do you mean here about the fecundity of deeper organisms? It is not clear how this fecundity contributes to the deep reef refugia hypothesis.

Line 286-287: The recovery of this species could potentially then rely on the recruitment of coral larvae

Lines 308-310: In this case, a molecular approach is required to correctly and quickly delineate coral species and provide a better understanding of coral species biology and ecology.

Lines 314-315: …to evaluate the contribution of shallow and deep populations to recruitment and population maintenance.

Line 315: While molecular approaches have been used



Figure 2:
It would be more useful if the color of the circles surrounding the clades was the same color as the writing. For me it looks like the writing “Type 8” is in brown but the circle around the haplotype is grey.

Table 1:
Summary of haplotype diversity per site, corresponding to literature references, and
their geographic locations previously collected.


It would also be nice to have the species name following Schmidt-Roach et al. (2014) in the table so that people less familiar with all the various types could more easily understand what species these haplotypes belong to.


I do not feel that Table 2 is needed because it presents the same data as Figure 2 but without the genetic distances.

---

## Round 0.2 · accepted · Accept

I am satisfied with the changes made to the manuscript.

#